# Development of a Fully Automated Desktop Analyzer and Ultrahigh Sensitivity Digital Immunoassay for SARS-CoV-2 Nucleocapsid Antigen Detection

**DOI:** 10.3390/biomedicines10092291

**Published:** 2022-09-15

**Authors:** Ryotaro Chiba, Kei Miyakawa, Kotaro Aoki, Takamitsu J. Morikawa, Yoshiki Moriizumi, Takuma Degawa, Yoshiyuki Arai, Osamu Segawa, Kengo Tanaka, Hideji Tajima, Susumu Arai, Hisatoshi Yoshinaga, Ryohei Tsukada, Akira Tani, Haruhito Fuji, Akinobu Sato, Yoshikazu Ishii, Kazuhiro Tateda, Akihide Ryo, Toru Yoshimura

**Affiliations:** 1Research and Development, Abbott Japan LLC, Matsudo 270-2214, Japan; 2Department of Microbiology, Yokohama City University School of Medicine, Yokohama 236-0004, Japan; 3Department of Microbiology and Infectious Diseases, Toho University School of Medicine, Tokyo 143-8540, Japan; 4Precision System Science Co., Ltd., Matsudo 271-0064, Japan; 5Sumitomo Bakelite Co., Ltd., Tokyo 140-0002, Japan; 6Olympus Corporation, Hachioji 192-8507, Japan

**Keywords:** desktop analyzer, digital ELISA, digital immunoassay, SARS-CoV-2, nucleocapsid antigen

## Abstract

Background: The severe acute respiratory syndrome coronavirus 2 (SARS-CoV-2) outbreak has had a significant impact on public health and the global economy. Several diagnostic tools are available for the detection of infectious diseases, with reverse transcription-polymerase chain reaction (RT-PCR) testing specifically recommended for viral RNA detection. However, this diagnostic method is costly, complex, and time-consuming. Although it does not have sufficient sensitivity, antigen detection by an immunoassay is an inexpensive and simpler alternative to RT-PCR. Here, we developed an ultrahigh sensitivity digital immunoassay (d-IA) for detecting SARS-CoV-2 nucleocapsid (N) protein as antigens using a fully automated desktop analyzer based on a digital enzyme-linked immunosorbent assay. Methods: We developed a fully automated d-IA desktop analyzer and measured the viral N protein as an antigen in nasopharyngeal (NP) swabs from patients with coronavirus disease. We studied nasopharyngeal swabs of 159 and 88 patients who were RT-PCR-negative and RT-PCR-positive, respectively. Results: The limit of detection of SARS-CoV-2 d-IA was 0.0043 pg/mL of N protein. The cutoff value was 0.029 pg/mL, with a negative RT-PCR distribution. The sensitivity of RT-PCR-positive specimens was estimated to be 94.3% (83/88). The assay time was 28 min. Conclusions: Our d-IA system, which includes a novel fully automated desktop analyzer, enabled detection of the SARS-CoV-2 N-protein with a comparable sensitivity to RT-PCR within 30 min. Thus, d-IA shows potential for SARS-CoV-2 detection across multiple diagnostic centers including small clinics, hospitals, airport quarantines, and clinical laboratories.

## 1. Introduction

A case of severe acute respiratory syndrome coronavirus-2 (SARS-CoV-2) infection, called coronavirus disease 2019 (COVID-19), was first reported in Wuhan, China at the end of 2019 [1]. This infectious disease caused a global pandemic [2], with over 300 million infections reported worldwide. The number of COVID-19-related deaths at the time of writing was 5 million [3].

COVID-19 is diagnosed through molecular testing of viral RNA. Reverse transcription-polymerase chain reaction (RT-PCR) testing for viral RNA is recommended to diagnose COVID-19 [4].

SARS-CoV-2 nucleocapsid (N) protein as an antigen detection by immunoassays has been developed as an alternative to PCR testing. The direct detection of SARS-CoV-2 antigens reflects its potential infectivity [5]. Some SARS-CoV-2 antigen tests with commercial, fully automated in vitro diagnostic devices have been approved by the US Food and Drug Administration [6] and the Japan Pharmaceuticals and Medical Devices Agency [7]. However, the sensitivity of current antigen testing is much lower than that of RT-PCR [6,8,9,10].

Digital enzyme-linked immunosorbent assay (ELISA) has recently been developed as an ultrahigh sensitivity immunoassay [11,12,13,14,15]. This assay uses a single-molecule detection technology with partitioned small water-in-oil droplets (femtoliter chamber) and a fluorogenic substrate, enabling the detection of a single enzyme using a simple optical system. The small volume of the femtoliter chamber (approximately 50 fL or less in our case), which is used as a reactor for the enzyme, enables the enclosed enzyme to produce a detectable signal by accumulating fluorescent reaction product molecules in a short time [11,12,13,14,15]. This technology has been expanded to ELISA and has been dubbed “digital ELISA” or “digital immunoassay (d-IA)”. The sensitivity of d-IA is 1000 times higher than that of the conventional ELISA method [16]. Therefore, a single-molecule array SARS-CoV-2 N-protein antigen test based on d-IA was approved by the US Food and Drug Administration for emergency use [17], with a limit of detection and percent positive agreement (PPA) with PCR of 0.02 pg/mL N-protein [18] and 97.7%, respectively [10,17]. Testing could be semi- or fully automated, using a fully automated system. Although testing using d-IA presents several advantages compared to RT-PCR testing, its usage is also limited to clinical laboratories because of the size of the instruments (141 × 79 × 161 cm). A single-molecule array only requires 80 min to produce the first result on a sample [19].

Lateral flow immunoassays are used to make on-site diagnostic decisions regarding COVID-19 at point-of-care testing [4,20]. The testing time associated with this technique can be as short as 15–30 min, owing to a simpler workflow compared to other diagnostic techniques. Point-of-care testing has some advantages, including on-site diagnosis in emergency departments and other healthcare facilities such as clinics. However, the corresponding PPA is 22.9–71.4% [21], and false negatives are a concern [22].

To overcome these problems, we developed an ultrasensitive SARS-CoV-2 antigen detection test using d-IA with a fully automated desktop analyzer. We targeted the N protein because it is expressed in large quantities in SARS-CoV-2 variants [23,24]. The size of our instrument was 32 × 60 × 57.5 cm, making it compact enough to place onto a desk. The assay time was less than 30 min, and the simple operation was similar to that of commercial fully automated in vitro diagnostic devices. We assessed SARS-CoV-2 d-IA using a desktop analyzer by measuring clinical samples as SARS-CoV-2 positive/negative nasopharyngeal (NP) swabs and found an ultrahigh sensitivity comparable to that of RT-PCR.

## 2. Materials and Methods

### 2.1. Specimens

A total of 247 NP swab specimens were collected from patients with confirmed and suspected COVID-19 at Toho University and Yokohama City University. RT-PCR confirmed that all the specimens were positive or negative. The specimens were stored at −80 °C. Furthermore, informed consent was obtained from all participants. The study protocol was approved by the ethics committee of the Faculty of Medicine of Toho University (no. A20028_A20020_A20014_A19099, approved on 8 May 2020) and the institutional review board of Yokohama City University (IRB no. B200800106, approved on 18 January 2021).

### 2.2. Fully Automated Desktop Analyzer

A fully automated desktop analyzer was manufactured by Precision System Science LLC (Figure 1A). The analyzer consisted of a heater unit to control the incubation temperature, eight pipetting units with disposable tips to aspirate/dispense liquid, a detection unit with a compact digital camera (TG-6 with custom firmware, Olympus, Shinjuku, Japan) (Figure 1B), an illumination system, a PC unit, and an electric equipment unit. Bead transfer and B/F separation were performed using Magtration^®^ technology [25]. The reagents used for the antigen assay were encapsulated in a microtiter cartridge. This system allowed batch processing with eight samples for testing.

### 2.3. Microwell Array Device (Digital Device)

A cyclic olefin polymer-based femtoliter chamber array, hereafter referred to as a digital device, was manufactured by Sumitomo-Bakelite (Figure 1C). The nominal dimensions of the microwell were 4 µm diameter, 3 µm depth, and 9 µm center-to-center distance. The thickness of the digital device was 0.5 mm. A total of 4.2 × 10^5^ microwells were present at the bottom of the 5.6 mm bore diameter (6.9 mm external diameter) well.

### 2.4. Reagents and Virus Preparation

Two anti-SARS-CoV-2 nucleocapsid protein antibodies were developed at Yokohama City University (Kanto Chemical, Tokyo, Japan) [26]. One antibody was coated onto Magnosphere™ MS300 carboxyl beads (JSR, Tokyo, Japan) with 1-ethyl-3-(e-dimethylaminopropyl) carbodiimide (Sigma-Aldrich, St. Louis, MO, USA) in MES buffer. Another antibody was conjugated to calf intestine alkaline phosphatase (BBI Solutions, Salisbury, UK) using trans-cyclooctene and tetrazine click chemistry (Click Chemistry Tools, Scottsdale, AZ, USA). The alkaline phosphatase-conjugated antibody was purified using a Superdex 200 Increase column (Cytiva, Uppsala, Sweden) in phosphate-buffered saline. Manually synthesized pyranine phosphate was used as a fluorogenic substrate for alkaline phosphatase [27]. In addition, recombinant SARS-CoV-2 nucleocapsid protein was obtained from Abbott Laboratories. Authentic SARS-CoV-2 viruses, Wuhan/A (WK-521, GISAID #EPI_ISL_408667), Alpha/B.1.1.7 (QHN001, GISAID #EEPI_ISL_804007), Beta/B.1.351 (TY8-612, GISAID #EPI_ISL_1123289), Gamma/P.1 (TY7-503, GISAID #EPI_ISL_877769), Kappa/B.1.617.1 (TY11-330, GISAID #EPI_ISL_2158613), and Delta/AY.122 (TY11-927, GISAID #EPI_ISL_2158617), were obtained from the National Institute of Infectious Diseases in Japan and handled in biosafety level 3 laboratories. Preparation of other viruses were described previously [26].

### 2.5. Antigen Assay

A schematic representation of the SARS-CoV-2 antigen d-IA is shown in Figure 2. NP swab samples were mixed with the same volume of pretreatment solution and incubated at 25 °C for 15 min to inactivate the virus. Then, 100 µL of the pretreated sample, 50 µL of anti-SARS-CoV-2 N protein antigen (1st Ab)-coated magnetic bead solution, 50 µL of assay specimen diluent, and 50 µL of alkaline phosphate-labeled anti-SARS-CoV-2 N protein antigen (2nd Ab) solution were incubated in a microtiter cartridge well at 37 °C for 8 min. After washing, the beads were mixed with 200 µmol/L pyranine phosphate in an alkaline buffer solution (substrate solution) introduced into the digital device. After the beads had settled with a magnet, 150 µL of fluorinated oil (Fluorinart FC-40) (3M, Maplewood, MN, USA) was added. Because the density of FC-40 was greater than that of water, the aqueous solution and oil phases were switched. Therefore, all microwells were sealed with FC-40 to form femtoliter chambers [28]. Then, a black dye solution was added to the top of the aqueous solution to reduce background fluorescence signals. The device was incubated at 37 °C for 5 min to allow for an enzymatic reaction with the conjugate. During incubation, bead and fluorescence images were captured sequentially from lane to lane with a compact digital camera under illumination through an emission filter (LV0510, Asahi Spectra, Tokyo, Japan) [29] at 55 s intervals. Bead images were captured under a 4× green light-emitted-diode (MLEGRN-A1-0000-000001, CREE, Durhan, NC, USA) for scattering images. Fluorescence images were obtained using a 4× blue light-emitted-diode (XQEROY-H0-0000-000000N01, CREE, Durhan, NC, USA) equipped with an excitation filter (SV0490, Asahi Spectra, Tokyo, Japan) to excite the pyranine. The neutralization test was performed using the same method as the d-IA, with a different pretreatment solution. The pretreatment solution for the neutralization test included 100 nM of anti-SARS-CoV-2 N protein antibody.

Lumipulse SARS-CoV-2 Ag (Fujirebio, Tokyo, Japan) was also used for SARS-CoV-2 Ag detection according to the manufacturer’s instruction.

### 2.6. Image Processing

The images were analyzed using Python software. A green channel image was isolated for analysis. For bright droplet counting, four time series of fluorescence microscopy images were aligned using an image registration module (pyStackReg), and linear regression was performed at each pixel to generate the slope and intercept images. The slope and intercept images reflect the enzyme active (i.e., time-dependent fluorescence change) and constant bright pixels, respectively. The bright droplets were subsequently detected using an appropriate global threshold. An average image from the four bead images was used with an appropriate intensity threshold for trapped bead counting. 

### 2.7. Data Analysis

The following equation was used to determine the fraction of bright droplets:$$\mathrm{Signal}\%=\frac{N_{b}D}{N_{t}B}\times 100\%$$
where N_b_D and N_t_B are the number of bright droplets with beads and beads trapped by femtoliter chambers, respectively.

### 2.8. Cross-Reactivity and Variant Detection

Samples of human coronaviruses (HCoV) and common respiratory viruses were prepared at Yokohama City University as previously described [26]. HCoV-OC43 and 229E were quantified using RT-PCR, as previously described [30]. Human rhinoviruses 14 and 16, as well as respiratory syncytial virus were quantified using a 50% tissue culture infectious dose (TCID50) assay. Influenza A viruses H1N1 and H3N2, as well as the influenza B virus Victoria and Yamagata lineages were quantified as previously described [31]. Viral samples were inactivated by the addition of NP-40 to a final concentration of 0.5% (*v*/*v*) immediately before each assay. The sample was diluted to the target concentration using a BD Universal Viral Transport Kit (BD Biosciences, San Jose, CA, USA). 

Each SARS-CoV-2 antigen assay was conducted as described above and repeated three times. Each variant of SARS-CoV-2 assay was also performed.

## 3. Results

### 3.1. Analytical Sensitivity

The analytical sensitivity of the SARS-CoV-2 d-IA was evaluated using recombinant SARS-CoV-2 N protein-diluted panels at 0, 0.01, 0.05, 0.1, 0.5, and 1 pg/mL (Figure 3). The limit of detection was determined by extrapolating the concentration at which the signal was equal to the background signal plus three standard deviations (SDs) of the background signal [32]. The calculated limit of detection was 0.0043 pg/mL, with a coefficient of variation (CV) of 14.2%.

### 3.2. Cutoff Setting and Clinical Specificity

Clinical specificity was evaluated using 159 specimens from RT-PCR-negative NP swabs for SARS-CoV-2 d-IA (Appendix A). Of the 159 specimens, 156 showed a signal% lower than 0.5%, whereas three specimens (ID: 2490, 2634, and 3237) showed a higher signal%. A neutralization test was conducted on the three specimens, and the signals were found to decrease (Appendix A). The mean and SD of the 156 specimens were 0.0347% and 0.0194%, respectively. The cutoff value was set to 0.228% (mean ± 10 SDs of 156 specimens). Using this cutoff, the specificity was estimated to be 98.1% (156/159). The specificity, upon exclusion of the three neutralized specimens, was estimated to be 100% (156/156).

### 3.3. Clinical Sensitivity

Clinical sensitivity was evaluated using 88 specimens from RT-PCR-positive NP swabs for SARS-CoV-2 d-IA (Figure 4). The sensitivities of the cutoff value were 94.3% (83/88) and 100% (58/58) with <35 Ct value and <30 of PCR results, respectively (Table 1). 

We measured a PCR-positive swab (8.31 × 10^7^ copies/mL) and the diluted samples using the digital assay, Lumipulse SARS-CoV-2 Ag and RT-PCR (Figure 5). The digital assay and the Lumipulse detected 870 copies/mL (Ct value, 39.6) and 7340 copies/mL (Ct value, 35.9) respectively.

### 3.4. Cross-Reactivity and Variant Detection

We assessed the cross-reactivity of related human coronaviruses with other respiratory viruses. Other respiratory viruses, such as HCoV-229E, HCoV-OC43, HCoV-NL63, rhinovirus 14, rhinovirus 16, influenza A H1N1, influenza A H3N2, influenza B Yamagata, influenza B Victoria, and respiratory syncytial viruses were not detected in the assay (Figure 6A). Five major circulating variants of SARS-CoV-2 (Alpha/B.1.1.7, Beta/B.1.351, Gamma/P.1, Kappa/B.1.617.1, and Delta/AY.122) were measured. All variants were detected as positive, and the signal% of the same concentration samples was equivalent to that of the original SARS-CoV-2 (Figure 6B). 

## 4. Discussion

In this study, we developed a fully automated desktop analyzer and ultrahigh sensitivity d-IA for SARS-CoV-2 detection. The limit of detection for SARS-CoV-2 N protein was 0.0043 pg/mL (Figure 3). We measured PCR-positive swabs, diluted samples, and subsequently detected 870 copies/mL (Figure 5). The analytical sensitivity of the SARS-CoV-2 N protein and SARS-CoV-2 was higher than that of other antigen tests, including the single-molecule array SARS-CoV-2 N-protein antigen test [9,18]. The Ct value of the diluted sample (870 copies/mL) was 39.6. Generally, a Ct value < 40 is an index of SARS-CoV-2 RNA positivity [33]. The assay showed the same analytical sensitivity as that of the diluted RT-PCR-positive samples. The specificity and sensitivity were 98.1% (156/159) and 94.3% (83/88), respectively (Table 1 and Appendix A). The assay results were highly concordant with RT-PCR results. Five RT-PCR-positive NP swabs showed lower signal% than the cutoff, and there was a possibility that these swabs were collected from patients at different phases of infection [34]. Three RT-PCR-negative NP swabs (ID: 2490, 2634, and 3237) showed a higher signal than the cutoff, which decreased in the neutralization test (Appendix A). This result suggests that NP swabs were positive for the virus N-protein antigen. The SARS-CoV-2 d-IA targets the SARS-CoV-2 N protein. The amount of N-protein was calculated to be 1000 times higher than that of the viral RNA molecule [23]. We assumed that a much higher amount of the virus N-protein existed in the nasopharynx of individuals infected with SARS-CoV-2 than the virus RNA molecule.

SARS-CoV-2 d-IA showed no cross-reactivity with related human coronaviruses, including SARS-CoV and other respiratory viruses (Figure 6A). We developed an immunoassay utilizing highly specific monoclonal antibodies for the detection of the SARS-CoV-2 N-protein antigen [26]. We performed the SARS-CoV-2 d-IA assay using five major circulating variants [4,35]. We found that the detection performance of the five variants was similar (Figure 6B). The sequence of the SARS-CoV-2 N-protein gene is highly conserved because mutation of the N-protein is critical [35]. Thus, the assay developed in this study can be used to detect a broad range of SARS-CoV-2 variants.

Our analysis was performed using a compact (32 × 60 × 57.5 cm) fully automated analyzer, which can be placed in small spaces such as a laboratory bench. This device can perform all steps of d-IA, including immunoreaction, digital detection, and analysis in less than 30 min. Commercial in vitro diagnostic devices also perform these steps in the same timeframe. However, their usage is commonly limited to hospitals or clinical laboratories because of the larger size of the analyzer. A fully automated d-IA analyzer has been developed [19]. However, it is large (141 × 79 × 161 cm) for use in general clinical laboratories, especially in small hospitals, emergency departments, or clinics. Moreover, the assay is time-consuming and requires more than 80 min. 

Point-of-care testing such as lateral flow immunoassay is an alternative COVID-19 on-site diagnostic tool to RT-PCR testing because it is quicker and less complex and utilizes more compact instruments [4]. However, false negatives and false positives caused by low sensitivity and specificity have been reported [22]. Our compact fully automated analyzer shows great potential for use in on-site testing of SARS-CoV-2, with the ability to provide results in less than 30 min. This device could provide a SARS-CoV-2 antigen test sensitivity comparable to that of real-time RT-PCR testing in an emergency department or other healthcare facilities. In particular, this d-IA can be used to monitor early SARS-CoV-2 infection in healthcare workers, or for entry screening of patients in hospitals or nursing homes.

Our analyzer was miniaturized as follows: (1) utilizing a simple “cup type” digital device (no flow channel, no housing) with a simple bead encapsulation method, (2) employing a commonly used compact digital camera and illumination system, and (3) using Magtration^®^ technology (no fluid line, no sample moving actuator). To accomplish (1), we introduced a protocol for the addition of black dye to the aqueous solution phase to switch the oil phase and to seal the microwells [36]. This dye considerably reduces unintended background and/or noise fluorescence but has little effect on the fluorescence signal from the reacted enzyme. For (2), we used a commercially available digital camera with sufficient performance to obtain images for d-IA. The camera had optics, sensors, and autofocus specifications that allowed us to integrate a new light source system with light emitted diodes on the side of the objective lens. The imaging setup was smaller than that of conventional scientific cameras and optics, allowing it to be used with cup-type devices. To accomplish (3), the Magtration^®^ instrument consists of simple apparatus such as a pipetting robot, a one-way-moving mechanical linear actuator for the pipette (or stage), and a magnet [26]. Therefore, the analyzer was compact and inexpensive. This device can be used in clinical laboratories, laboratories with limited space, and small hospitals and clinics to facilitate the performance of ultrasensitive assays.

In case of more than 10^5^ copies/mL, the signal% of d-IA showed saturation (Figure 5). It was due to the limitation of digital detection. All of the assay beads applied to the high-concentration sample were associated with enzyme-labeled antibodies. In addition, the output of image analysis algorithm we developed became variable with a high-intensity image from the high-concentration sample. Therefore, further improvement in the algorithm to expand the dynamic range [37] would be needed to overcome this limitation.

The SARS-CoV-2 d-IA developed in this study, using the novel compact fully automated analyzer, mirrored the high performance and sensitivity of RT-PCR. Further studies are needed to establish commercial in vitro diagnostic assays that consider stability and specimen evaluation data, including positive and negative specimens.

## 5. Conclusions

In conclusion, the fully automated desktop analyzer is an ultrahigh sensitivity immunoassay system with a sensitivity comparable to that of RT-PCR. Its space-saving properties, simplicity, and fast throughput rate allow it to require less effort to detect SARS-CoV-2.

## Figures and Tables

**Figure 1 biomedicines-10-02291-f001:**
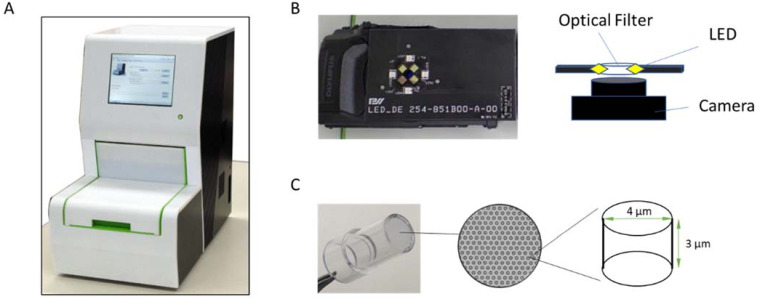
The desktop analyzer and digital device. (**A**) The desktop analyzer. (**B**) Illumination and detection system equipped with a desktop analyzer. (**C**) The digital device, micro-well, and image of the micro-well.

**Figure 2 biomedicines-10-02291-f002:**
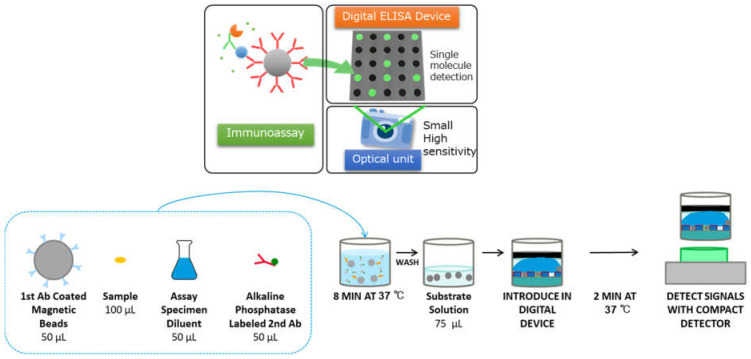
A schematic drawing of SARS-CoV-2 Ag digital immunoassay.

**Figure 3 biomedicines-10-02291-f003:**
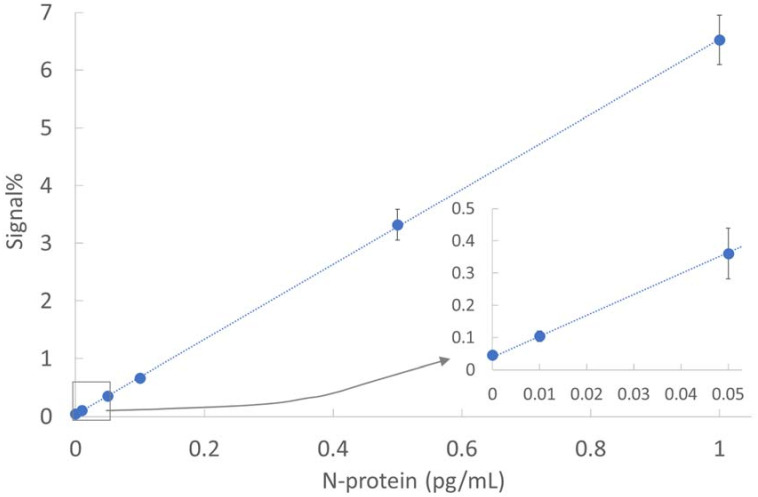
Dose response with recombinant SARS-CoV-2 nucleocapsid protein.

**Figure 4 biomedicines-10-02291-f004:**
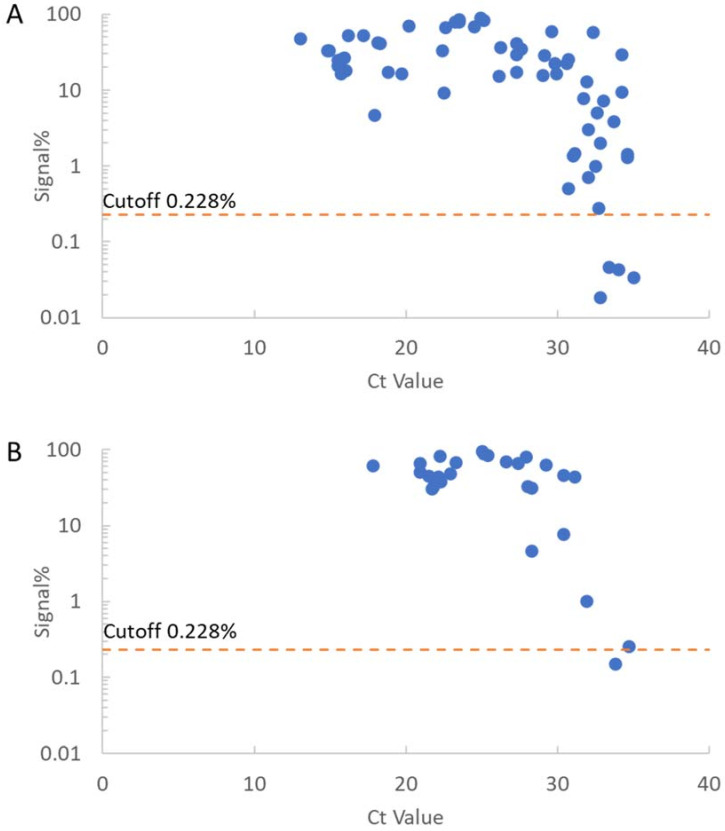
Measurement of RT-PCR-positive specimen and dose response with RT-PCR-positive specimen. (**A**) Results of 61 RT-PCR-positive NP swab specimens from Toho University. (**B**) Results of 27 RT-PCR-positive NP swab specimens from Yokohama City University.

**Figure 5 biomedicines-10-02291-f005:**
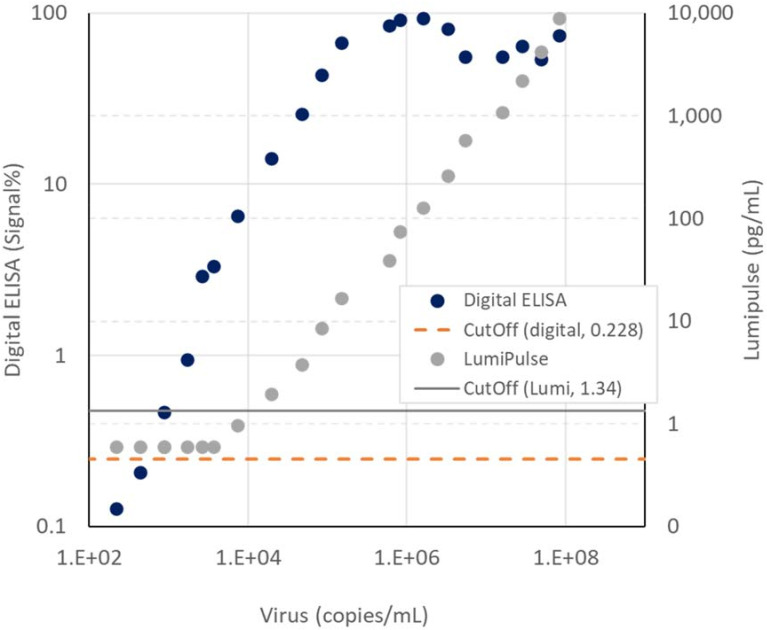
Results of the measurement of 0, 217, 434, 869, 1708, 2634, 3367, 7335, 1.96 × 10^4^, 4.77 × 10^4^, 8.42 × 10^4^, 1.49 × 10^5^, 5.97 × 10^5^, 8.14 × 10^5^, 1.61 × 10^6^, 3.32 × 10^6^, 5.45 × 10^6^, 1.58 × 10^7^, 2.85 × 10^7^, 4.90 × 10^8^, and 8.31 × 10^8^ copies/mL diluted samples. Cutoff: 0.228% for d-IA and 1.34 for Lumipulse.

**Figure 6 biomedicines-10-02291-f006:**
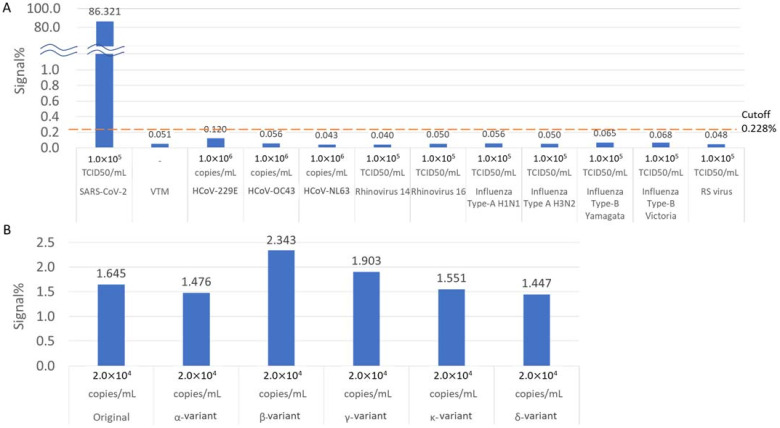
Cross-reactivity and variant detection. (**A**) Results of 1.0 × 10^5^ TCID50/mlL SARS-CoV-2, 1.0 × 10^6^ copies/mL HCoV-229E, 1 × 10^6^ copies/mL HCoV-OC43, 1.0 × 10^6^ copies/mL HCoV-NL63, 1.0 × 10^5^ TCID50/mL Rhinovirus 14, 1.0 × 10^5^ TCID50/mL Rhinovirus 16, 1.0 × 10^5^ TCID50/mL Influenza Type-A H1N1, 1.0 × 10^5^ TCID50/mL, Influenza Type A H3N2, 1.0 × 10^5^ TCID50/mL, 1.0 × 10^5^ TCID50/mL Influenza Type B Yamagata, 1.0 × 10^5^ TCID50/mL Influenza Type B Victoria, and 1.0 × 10^5^ TCID50/mL respiratory syncytial RS virus. VTM, Viral Transport Medium. (**B**) Result of 2 × 10^4^ copies/mL SARS-CoV-2 α, β, γ, κ, and δ variants.

**Table 1 biomedicines-10-02291-t001:** Percent positive agreement with RT-PCR.

PCR Ct	PPA with RT-PCR
<35	94.3% (83/88)
<30	100.0% (58/58)

## Data Availability

Not applicable.

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
