# Peer review of "Development of a Fully Automated Desktop Analyzer and Ultrahigh Sensitivity Digital Immunoassay for SARS-CoV-2 Nucleocapsid Antigen Detection"

_biomedicines, 2022, doi:10.3390/biomedicines10092291_

Round 1
Reviewer 1 Report
This paper presents a digital immunoassay platform that enables high-sensitivity detection of SARS-CoV-2 nucleocapsid (N) protein. The authors demonstrate digital ELISA measurement of N protein in nasopharyngeal swabs from patients using a fully automated desktop device. The technology achieved the measurement with a very low limit of detection and a short assay turnaround. The analysis of the obtained data appears solid and supports the authors' claim on assay performance. The paper generally presents high-quality work with rigorous validation of the platform and can be recommended for publication in Biomedicines. But the authors are encouraged to address the following relatively minor issues in their revised manuscript before its publication:
(1) The rationale for developing the digital immunoassay method with ultra-high sensitivity comparable to that of RT-PCR is not so obvious. Therefore, the authors should provide more in-depth discussions on issues in COVID-19 diagnostics by current lateral flow immunoassays with the limited sensitivity, merits of the digital immunoassay relative to cost-effective lateral flow immunoassays and other existing digital immunoassay platforms (e.g. Simoa in ref [20]), and impacts of the digital immunoassay on future COVID-19 diagnostics.
(2) In Figure 5, notable signal saturation and drop in the digital ELISA signal exist for the virus concentration > 10^6 copies/mL. Why? Did the system reach the limit of the Poisson statistics regime ensuring single-molecule detection?
(3) In the discussion and conclusion sections, the authors claim that the presented digital immunoassay platform allows for SARS-CoV-2 detection with less effort and financial resources. But they should estimate the instrument and assay running costs of the proposed platform and compare them to those of other methods (e.g., lateral flow immunoassay kits and Simoa) to provide evidence for their claim.
Reviewer 2 Report
1. Title - does not seem to be hyphenated correctly at the end of each line.
2. Title and Abstract - in the title, the digital immunoassay is described as "ultrahigh" sensitivity, but in the abstract, it is called "highly" sensitive. Are there definitions or standards that distinguish "ultrahigh" and "highly" sensitive? If there are, those standards should be explained. If there are not, the authors should pick a description of the sensitivity and use it consistently throughout the manuscript.
3. The limit of detection is presented to three significant digits, although the %CV is over 14%. This does not seem appropriate. I would recommend limiting the LOD to two significant digits, unless there is some justification to using more than two.
4. In the abstract, line 38 states the digital immunoassay has "equivalent" sensitivity to RT-PCR. That may be too strong of a claim. "Comparable" sensitivity might be a better description.
5. Many of the limitations of RT-PCR based testing described in the introduction (lines 52 to 55) are simply inaccurate. Even before COVID, there were rapid, small, affordable nucleic acid amplification-based devices used in point of care settings in the US (and other developed countries) for detection of flu, group A strep, RSV, and other infectious disease. Results could be obtained in 20 minutes on some of these instruments. What the authors are describing is not the current state.
6. In figure 5, the dip in Digital ELISA(Signal%) in the range of about 10^7 viral copies/mL is not explained. Is this the hook effect, or something else?
Overall, this is well written and the technology is nicely described. It should be of significant interest to readers.
